# Neurocognitive Endophenotypes for Eating Disorders: A Preliminary High-Risk Family Study

**DOI:** 10.3390/brainsci13010099

**Published:** 2023-01-04

**Authors:** Edoardo Pappaianni, Manuela Barona, Gaelle E. Doucet, Christopher Clark, Sophia Frangou, Nadia Micali

**Affiliations:** 1Mental Health Services in the Capital Region of Denmark, Eating Disorders Research Unit, Psychiatric Center Ballerup, 2750 Ballerup, Denmark; 2Great Ormond Street Institute of Child Health, University College London, London WC1N 1EH, UK; 3Institute for Human Neuroscience, Boys Town National Research Hospital, Omaha, NE 68010, USA; 4Department of Psychiatry, Icahn School of Medicine at Mount Sinai, New York, NY 10029, USA; 5Department of Psychiatry, University of British Columbia, Vancouver, BC V6T 1Z3, Canada

**Keywords:** eating disorders, high-risk studies, executive function, resting-state fMRI, endophenotypes, familial high-risk

## Abstract

Eating disorders (EDs) are psychiatric disorders with a neurobiological basis. ED-specific neuropsychological and brain characteristics have been identified, but often in individuals in the acute phase or recovered from EDs, precluding an understanding of whether they are correlates and scars of EDs vs. predisposing factors. Although familial high-risk (FHR) studies are available across other disorders, this study design has not been used in EDs. We carried out the first FMH study in EDs, investigating healthy offspring of women with EDs and controls. We preliminarily aimed to investigate ED-related neurocognitive and brain markers that could point to predisposing factors for ED. Sixteen girls at FHR for EDs and twenty control girls (age range: 8–15), completed neuropsychological tests assessing executive functions. Girls also underwent a resting-state fMRI scan to quantify functional connectivity (FC) within resting-state networks. Girls at FHR for EDs performed worse on a cognitive flexibility task compared with controls (F = 5.53, *p* = 0.02). Moreover, they showed different FC compared with controls in several resting-state networks (*p* < 0.05 FDR-corrected). Differences identified in cognitive flexibility and in FC are in line with those identified in individuals with EDs, strongly pointing to a role as potential endophenotypes of EDs.

## 1. Introduction

Eating disorders (EDs) such as anorexia nervosa (AN), bulimia nervosa (BN) and binge eating disorder (BED) are severe psychiatric disorders and have important consequences on physical health and psychosocial functioning [1]. EDs affect 10–20% of adolescents and young adults [2,3].

Research shows that individuals with EDs differ in terms of cognitive processing and brain structure and function compared with controls [4,5,6]. Although specific correlates have been identified in each diagnostic group, main neurocognitive alterations have been found across executive functions [7]. One of the most studied cognitive components in EDs, particularly in AN, is set-shifting. Difficulties in set-shifting (or cognitive flexibility, which refers to the ability to shift thoughts or actions according to situations) have obtained evidence as a correlate of EDs (although most of the literature focuses on AN) [8,9,10,11,12]. In addition, differences in other high-demand cognitive functions including working memory and sustained attention have been highlighted in relation to EDs [13].

From a functional perspective, several neuroimaging studies with ED patients have shown abnormal patterns of brain functional connectivity (FC) at rest that parallel these findings. Evidence highlights ED-specific abnormal FC during resting state within the default-mode (DMN) network. As its name suggests, the DMN consists of a set of areas that are active while not performing any particular task, i.e., in the complete state of rest [14,15,16]. This circuit is a large-scale network that includes regions such as the medial prefrontal cortex, precuneus and angular gyrus. It is involved in different cognitive functions, such as self-reflection, mind-wandering, self-referential mechanisms, autobiographical memory, self-evaluation processes [17,18,19], and it has frequently been seen to be involved in EDs [20,21]. However, other resting-state networks also seem to be affected by EDs, such as central executive (CEN) [20,21], ventral visual (VN), and sensorimotor (SMN) networks [22,23], the lateral visual and auditory networks [24] and the salience network [20,21]. These FC abnormalities mostly mirror cognitive and behavioral characteristics identified in EDs [25,26].

Although these findings are extremely useful in paving the way for an understanding of the neurobiology of EDs, they cannot help us disentangle whether characteristics identified are correlates/scars or endophenotypes (i.e., neurobehavioral traits that index genetic susceptibility for a psychiatric disorder [27] of EDs. Most studies, in fact, have focused on ED patients in an acute phase, or recovered from illness; hence, parsing out whether these alterations are secondary to active ED symptoms (e.g., starvation) or a scar of disorder is very difficult.

EDs show a strong genetic component [28]; they run in families, and studies have revealed this familial aggregation is likely to be due to genetic factors [29]. Familial high-risk (FHR) studies focus on individuals who have a biological relative, most commonly a parent, with a severe mental illness and who are at higher risk for the development of the disease. They rely on investigating subjects prior to the onset of the disorder and allow a prospective study of emerging psychopathology. This type of design allows identification of behavioral and biological predisposing factors, or traits of a mental disorder, without the confounding role of acute illness. The FHR design has been used across several psychiatric conditions [30], such as depression, bipolar disorder, and schizophrenia, to identify biological markers of mental illness. Studying individuals at FHR for EDs may reveal new information on predisposing neurobiological traits or endophenotypes for EDs. To date, only our previous cohort studies [31,32] have investigated cognition and neuropsychology in offspring of women with EDs, but an FHR study including both neuropsychology and neuroimaging has not been yet carried out. Therefore, capitalizing on the strength of the FHR design, we aimed to conduct a preliminary investigation of cognitive and neural markers in offspring of mothers with EDs compared with children born to women with no psychiatric disorders (control group). We focused our study on female offspring, given that ED prevalence is much higher in women than men [33]; thus, females are at higher risk of EDs. We aimed to test the hypothesis that children at FHR for EDs show behavioral neurocognitive differences compared with controls in line with those typical of EDs (specifically related to set shifting and related high-demanding cognitive components such as working memory and attention); and hypothesized that resting-state FC would mirror these differences. These characteristics, if present in those at FHR for EDs, may be serious candidates as ED endophenotypes.

## 2. Materials and Methods

### 2.1. Participants

All participants were recruited in the United Kingdom from ED clinics, ED support groups, and online adverts. In total, 36 girls and their mothers were recruited. The group at FHR included healthy daughters (N = 16, age-range:8–15 years, mean age:12.00 ± 2.19) of mothers with a history of current or past EDs (14 AN and 2 BN), while the control sample included healthy daughters (N = 20, age-range:8–15 years, mean age = 12.25 ± 1.94) of mothers with no history of EDs or any other psychiatric disorders (Table 1). Two participants from the control group dropped out before the MRI examination, reducing the control group sample to eighteen participants. The study was conducted in accordance with the Declaration of Helsinki and approved by the Ethical Committee of UCL and National Health Service. Informed consent was obtained from all subjects involved in the study. Both mothers and children were fully informed of the study, and both were required to give consent. Separate information sheets tailored to the child’s age (8 to 10 and 11 to 15 years old) were provided. Parents were able to decline participation on behalf of the child.

### 2.2. Experimental Procedures

We used the Structured Clinical Interview for Axis I DSM-IV Disorders (SCID-I) [34] to determine diagnoses of EDs. This tool is a valuable semi-structured diagnostic interview used to establish Axis I DSM-5 disorders [33]. Its ED module includes some open-ended questions on the presence of ED symptoms and cognitions. Relevant questions were repeated at appropriate times to determine the lifetime presence of ED symptoms.

Psychopathology (including ED symptoms) was assessed in offspring (FHR and HC) using the Strength and Difficulties Questionnaire (SDQ) [35] and the Development and Wellbeing Assessment (DAWBA) [36]. An MRI safety questionnaire was filled out before scanning to check for suitability for MRI examination. All experimental measures and demographic data were collected at Great Ormond Street Institute of Child Health, UCL.

### 2.3. Measures

#### 2.3.1. Socio Demographic Data

Maternal age, relationship status (married or cohabitating vs. not), education (up to O level/GCSE equivalent vs. A level and above) and ethnicity (White vs. non-White) were collected using an ad hoc questionnaire.

#### 2.3.2. Anthropometry and Pubertal Stage

Child’s weight and height were measured objectively in light clothes prior to the scanning session. Pubertal development was assessed with a pictogram containing a set of scaled drawings depicting external sex characteristics such as the size of breasts and the development of pubic hair presented to mothers [37].

#### 2.3.3. Child Psychopathology

We assessed child psychopathology using the DAWBA [36]. The DAWBA is one of the most commonly used tools for the identification of mental health problems in youths [38].

Moreover, the SDQ [35] was used as an additional screening tool. The SDQ is a brief behavioral questionnaire that assesses children and adolescents’ mental health on different subscales, such as emotional difficulties, conduct problems, hyperactivity problems, peer relationship difficulties and prosocial problems. It returns a score for each of these components, in addition to a total score.

#### 2.3.4. Global Intelligence

Intelligence was assessed using the Wechsler Abbreviated Scale of Intelligence II (WASI-II) [39]. The WASI includes four individual tests that assess full-scale IQ (FIQ).

#### 2.3.5. Neurocognitive Function

Neurocognitive functions were assessed using the computerized Cambridge Neuropsychological Test Automated Battery (CANTAB, www.cambridgecognition.com, accessed on 1 January 2019). The Attention Switching Task (AST) measures cognitive flexibility, Rapid Visual Information Processing (RVP) assesses sustained attention, and the Spatial Working Memory task (SWM) evaluates working memory.

##### Attention Switching Task (AST)

The AST assesses the ability to switch attention between arrow positions and directions, ignoring irrelevant information in the face of a distractor. Arrows appear on the left or right side of the screen and can point left or right. A cue at the top of the screen indicates whether the participant should press left or right, depending on the direction or position of the arrow. Some trials are position–direction-congruent (e.g., the arrow on the right side of the screen points to the right), while others are discordant and require higher cognitive demands. We recorded measures of performance (i.e., number of correct trials), response time, switching cost (i.e., difference in response between blocks with switching rules and blocks with rules remaining constant) and congruency cost (a difference in reaction time between congruent and incongruent condition).

##### Rapid Visual Information Processing (RVP)

RVP is a sensitive task to assess sustained attention. Numbers 2 to 9 are presented in pseudorandom order at a rate of 100 digits per minute. Participants are asked to recognize a target sequence of digits (e.g., 2-4-6, 3-5-7). We evaluated error measures and strategies (use of heuristic search sequences to maximize performance in the task).

##### Spatial Working Memory (SWM)

The SWM task is a measure of sustained attention and working memory where participants are asked to keep and manipulate visuospatial information. An increasing number of boxes is shown in each trial, and participants are asked to find the one that contains a target token through a process of elimination (i.e., the token cannot be in the same box twice). We recorded measures of errors and strategy to accomplish the task.

### 2.4. MRI Session

Participants were scanned on a Siemens Avanto 3T clinical MRI system at Great Ormond Street Institute of Child Health, UCL using a 64-channel head coil.

A T1-weighted 3D MPRAGE (Magnetization Prepared Rapid Acquisition Gradient Echo) scan was acquired with repetition time (TR) = 2300 ms; echo time (TE) = 2.74 ms; voxel size = 1 × 1 × 1 mm^3^; field of view = 256 mm; slice thickness = 1 mm; flip angle = 8°.

Resting-state fMRI data were collected using multi-band acceleration factor = 2; volumes = 300; TR = 1240 ms; TE = 26 ms; voxel size = 2.5 × 2.5 × 2.5 mm; slices = 40; slice thickness = 2.5 mm; field of view = 200 mm; flip angle = 75°. Resting-state acquisition time lasted approximately six minutes.

### 2.5. Data Analysis

#### 2.5.1. Demographic Data and Neurocognitive Function

All variables were examined individually to check for inconsistencies/outliers and their distribution. Demographic variables and covariates were compared across groups using chi-square (for categorical variables) and *t*-tests (for continuous variables).

In order to estimate neurocognitive differences, all output scores were included as dependent variables in an analysis of covariance (ANCOVA), with group as a fixed factor and age and FIQ as covariates. F-value, *p*-value, and eta-squared (η^2^) as measures of effect size are reported for each analysis.

Exploratory post hoc analyses were performed to study the effect of maternal diagnosis (AN vs. BN) on our results by stratifying analyses by maternal diagnosis.

All statistical analyses on neurocognitive outcomes were carried out using JASP (https://jasp-stats.org/,v.0.11, accessed on 1 January 2019). Statistical significance threshold was set at *p* ≤ 0.05 for all analyses.

#### 2.5.2. Resting-State fMRI Analysis

Preprocessing of the resting-state fMRI data were performed using Statistical Parametric Mapping v.12 (SPM12, https://www.fil.ion.ucl.ac.uk/spm/, UCL, London, UK) software and a wavelet-based method accessed on 1 January 2019 [40]. Preprocessing standard steps included slice timing correction, re-alignment, co-registration of functional images to the anatomical T1-weighted images followed by the spatial normalization to an unbiased standard template for pediatric data [41]. Based on the study population’s age range, we chose the brain template associated with the closest age range (age range: 7.5–13.5 years old) (http://www.bic.mni.mcgill.ca/ServicesAtlases/NIHPD-obj1#download, accessed on 1 January 2019. McConnell Brain Imaging Center, University St Montreal, Canada), accessed on 1 January 2019. The next step included spatial smoothing with a Gaussian kernel with a full-width half-maximum of 6 mm. We then applied a wavelet despiking technique for denoising signal transients related to head movements [40]. Participants showing head motion of more than 3 mm of displacement and 0.5 degrees of rotation in any direction were excluded. Based on these exclusion criteria, we excluded one FHR offspring and one control from further analyses. Neither the mean nor the maximal head motion differed between the two groups (*p* > 0.05 for both; average mean/maximal head motion for the control group: 0.03/0.72; average mean/maximal head motion for the patient group: 0.03/0.36).

Spatial probabilistic independent component analysis (ICA, temporal concatenation method) on FMRIB software library (FSL) 3.14 Melodic ICA software (FMRIB, Oxford, UK) [42] was used to identify resting-state networks across all participants. Preprocessed images from each participant (32 inputs) were entered into the ICA. High-pass temporal filtering was applied to exclude signal with a frequency under 0.01 Hz. We used the Laplace approximation to estimate the number of components. The output resulted in 40 independent components (ICs) common for the entire group of participants. Each IC was associated with a Z-map and a time-series. Each map was thresholded at a posterior probability threshold of *p* = 0.5, using an alternative hypothesis-testing approach based on the fit of a Gaussian/Gamma mixture model [43]. In a second step, based on these group ICs, we applied a dual regression approach to characterize each IC in each participant, through a Z-map and an individual time-series [44]. Independent components that included non-grey-matter regions, cerebellum, or were associated with high frequency signal (>0.1 Hz) were excluded from further analyses. We then identified the components covering the known resting-state networks, such as the DMN (n = 4), the CEN (n = 2), the SMN (n = 2), and the VN (n = 2) (Figure 1). The group ICA revealed four components covering the regions defined as part of the DMN [45] (Figure 1). Two components covered the posterior parts of the DMN with the IC6 including the posterior cingulate cortex/retrosplenial cortex and angular gyri and the IC35 mostly including the precuneus/posterior cingulate cortex. Two other components covered the anterior and dorsal parts of the DMN: the IC11 mostly included the anterior cingulate cortex, while the IC28 was mostly located in the dorsal medial prefrontal cortex. Four other components covered regions involved in central executive function such as the right and left working memory networks (IC3 and IC5, covering the lateral parietal–frontal cortex [46]); the dorsal attentional network (DAN, IC9, covering the intraparietal sulcus and frontal eye field [47]) and the salience network (IC20, covering the anterior insulas and dorsal anterior cingulate cortex [48]). Two components covered the dorsal and ventral regions of the SMN (IC4 and IC30, respectively). Lastly, two components covered the medial and lateral regions of the VN (IC1 and IC12, respectively).

After resting-state network extraction, second-level analyses were performed in order to determine spatial extent differences for each network between the groups. Individual Z-maps were entered into a second-level random-effects analyses separately for each network. Each comparison was restricted to the positive voxels of the network of interest (*p* < 0.001). We reported significant spatial differences at *p* < 0.001 (uncorrected, T > 3.4) with a cluster-level threshold of *p* < 0.05 (FDR corrected, corresponding to a spatial extent of minimum 20 voxels, approximately). All analyses were adjusted for age and pubertal development.

## 3. Results

### 3.1. Demographic Information

Children did not differ in age, pubertal development, general intelligence (FIQ). Maternal age, education and ethnicity were comparable across groups (see Table 1 for details). Fourteen women received a lifetime diagnosis of AN, 2 of BN.

### 3.2. Neurocognitive Function

All neurocognitive results are available in Table 2. We found a statistically significant effect of group (F(1,32) = 5.53, *p* = 0.02) on the AST in regard to the switching cost index, adjusting for age and FIQ, with a large effect size. Children at FHR for EDs had higher scores (i.e., they were slower in switching rule) compared with controls. No other statistically significant difference was detected in relation to spatial working memory and sustained attention (all *p*-values > 0.05), although there was a trend in girls at FHR for EDs having higher scores on SWM strategy (i.e., better use of a strategy to succeed in the SWM task) compared with controls, with a medium effect size.

In post hoc analyses, girls at FHR for AN showed higher switching cost scores in the AST compared with HC (F(1,30) = 6.86, *p* = 0.01), correcting for age and FIQ, confirming findings highlighted in the larger group.

### 3.3. Resting-State fMRI

Independent components covering the resting-state networks are shown in Figure 1.

We found spatial differences between groups in component parts of each of the four major networks (DMN, SMN, VN and CEN; all *p* < 0.05 FDR-corrected at the cluster-level). The posterior DMN (IC6) showed a reduction in spatial extent in two clusters located in the angular and middle temporal gyri in children at FHR for ED, while the dorsal DMN (IC28) showed reduction in the medial superior frontal gyrus. The same pattern was seen in the ventral SMN (IC30) and in two clusters located in the postcentral and inferior temporal gyri of the DAN (IC9).

Lastly, two clusters of the medial VN (IC1), corresponding to the fusiform and superior occipital gyri, showed increased activity in offspring at FHR for EDs relative to controls. Group comparisons are presented in Table 3 and Figure 2.

## 4. Discussion

This preliminary study is the first to investigate cognitive and neural markers of ED in FHR offspring. We compared cognitive and neural characteristics in girls at FHR for EDs (i.e., born to mothers with a past or current ED) and control girls (born to mothers with no history of EDs or other psychiatric disorders). Girls at FHR for EDs performed worse than controls in an attentional switching task, and differences between groups emerged in FC among resting-state networks. Girls at FHR showed decreased FC in the DMN, the SMN, and the DAN, whereas increased FC in the VN compared with controls.

Our results suggest that children at FHR for EDs present cognitive rigidity when switching, similar to what is seen in individuals with EDs, particularly AN [49]. Cognitive switching refers to the ability to switch attention from one mental state/task to another [49], and it is a core executive function. Difficulties in set-shifting in ED have been widely reported in the literature, although they seem clearer for AN than BN [12,50,51]. Results from our post hoc analyses are in line with these data, showing that girls at FHR for AN had significant difficulty in set-shifting compared with controls. Studies in healthy sisters or mothers of individuals with ED have shown mixed results. Healthy sisters of individuals with ED, mostly AN, but also BN and ED in general, have difficulties in cognitive switching compared with their ill sisters [52,53,54,55]. However, other evidence has shown normal set shifting in unaffected sisters of women with AN [54,56,57]. Our preliminary findings suggest a deficit in cognitive flexibility as a strong candidate as an endophenotype of ED, particularly AN.

Evidence of an impact of ED on working memory shows conflicting results [58,59]. In a community-based study, we showed increased performance in working memory in children of mothers with EDs [31]. Although we found a trend in the same direction, we did not replicate these findings in the current study.

Offspring at FHR showed decreased FC in the DMN, the DAN, and the SMN, as well as increased FC in the medial VN. In our study, children at high risk for EDs showed decreased FC within the DMN posterior areas (the angular and middle temporal gyrus), and the medial superior frontal gyrus. Such posterior regions play an important role in converging multisensory information and reorienting attention, manipulating mental states and social-affective dimensions [60,61]. Abnormal activation in these areas have been reported in adults [20] and adolescents [62] with AN.

The superior frontal gyrus is known to be related to self-awareness [63]. Aberrant connectivity in these areas could be related to body image distortion, a common feature of ED [5,64]. Notably, girls at FHR for EDs also showed differences in FC in the medial VN, specifically in regions involved in face and object recognition [65] such as the right fusiform gyrus and the left superior occipital gyrus. In a speculative fashion, we might consider this as a unique system deputed to body image processing, characterized by an abnormal processing of basic visual information (visual areas) reflected in the body-image distortion processed by high-level structures (frontal regions). Our findings might point towards this being a neural marker of distorted body image perception present in girls at FHR for EDs.

Decreased FC in the SMN has been shown in patients with BN [66], as well as in patients with restrictive AN [24]. Altered FC in this region may reflect abnormal patterns of inhibitory control and response in ED patients [67,68,69]. The presence of decreased FC in this network in children at FHR warrants the need to investigate its role as a neural endophenotype of EDs, as well as the reduced FC in the DAN, a network involved in voluntary orienting attention [70].

### Strengths and Limitations

One of the strengths of this study is its design, i.e., the first study using the FHR design in healthy children with no ED symptoms. This allows for measurements of potential endophenotypes that are less biased by confounding factors such as illness state. An intrinsic strength of this study is the innovation of both a neurocognitive and a neural investigation, this allows a broader and deeper understanding of potential markers of high risks for EDs.

This study has some limitations. First, given its preliminary nature, our sample size is small. Given the sample size, although all appropriate statistical steps were taken, we cannot exclude false negatives due to low statistical power. Larger studies are necessary to confirm our findings. We are currently carrying out a larger study of children at FHR for EDs, aimed at extending our findings. Secondly, in the same experimental group, we included offspring of mothers with AN and BN, based on many transdiagnostic theories that suggest similar pathophysiology for these disorders [71,72]. Future studies will aim to investigate FHR for each diagnostic group by including male population as well, that has been largely neglected in this research field so far.

## 5. Conclusions

This study highlights a difficulty in cognitive flexibility as a potential endophenotype of ED (particularly AN). We were able to also identify abnormal FC in the DMN, the SMN, the DAN, and the VN as neural correlates of FHR status for EDs. Taken together, such differences in FC parallel altered perception and processing of body image, abnormal attentional orienting, set-shifting, and inhibition control seen in patients with Eds. These findings might apply mostly to FHR status for AN given the diagnostic split of our sample.

## Figures and Tables

**Figure 1 brainsci-13-00099-f001:**
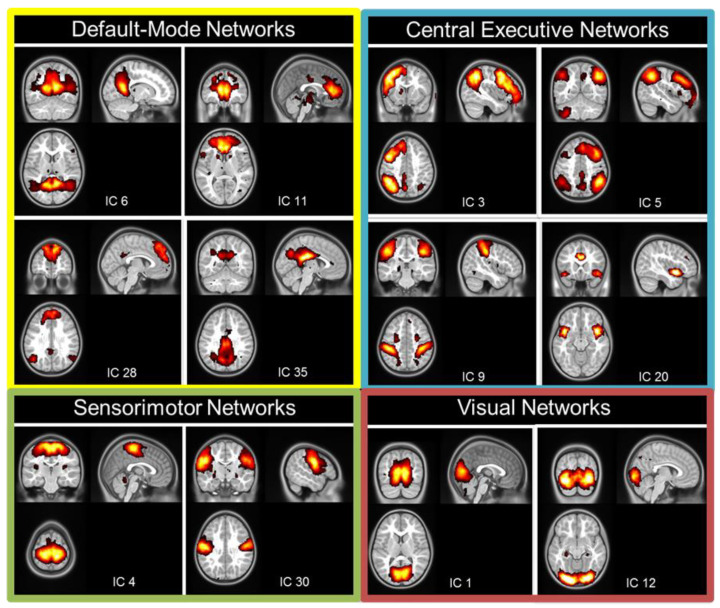
Spatial maps of the selected resting-state networks.

**Figure 2 brainsci-13-00099-f002:**
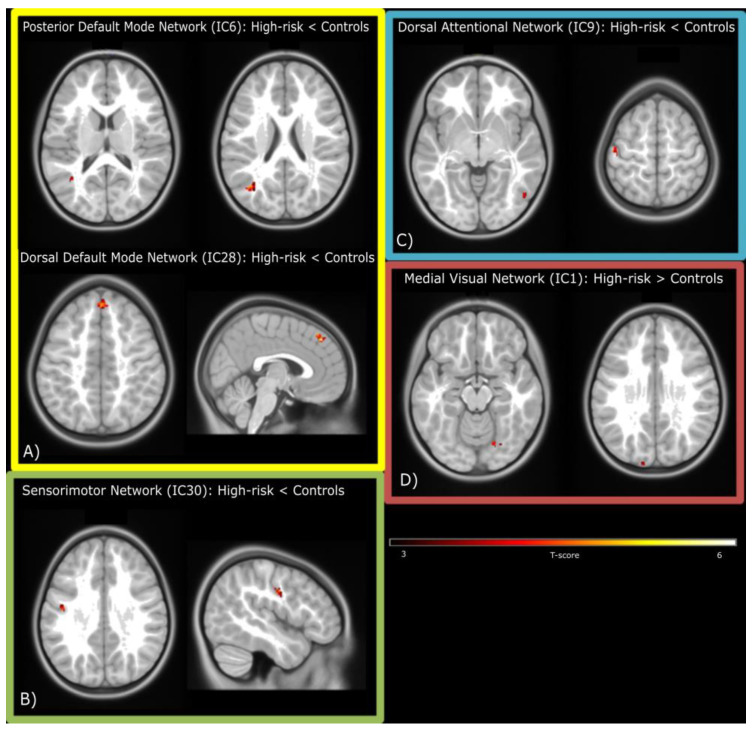
Spatial differences in resting-state networks between children at high risk for ED and controls. (**A**) Default-Mode Network; (**B**) Sensorimotor Network; (**C**) Dorsal Attentional Network; (**D**) Medial Visual Network. Contrasts are reported in the figure.

**Table 1 brainsci-13-00099-t001:** Demographics characteristics of participants.

	FHR for ED (N = 16)	HC (N = 20)	Group Comparison ^1^
	Means (SD)	Means (SD)	
Age (years)	12 (2.19)	12.25 (1.94)	t = −0.36, *p* = 0.72
Breast Developmental Stage	3 (1.55)	3.05 (1.43)	t = −0.10, *p* = 0.92
Pubic Hair Developmental Stage	3.25 (1.77)	3.16 (1.46)	t = 0.17, *p* = 0.87
WASI-Full-scale IQ (FIQ)	120.94 (12.04)	120.55 (14.45)	t = 0.08, *p* = 0.93
SDQ—Total difficulties	11 (4.87)	8.89 (5.85)	t = 1.14, *p* = 0.26
SQD—Emotional difficulties	3.8 (2.30)	3 (2.49)	t = 0.97, *p* = 0.34
SDQ—Conduct difficulties	1.60 (1.24)	1.21 (1.40)	t = 0.86, *p* = 0.40
SDQ—Hyperactivity difficulties	3.5 (2.59)	2.68 (2)	t = 1.13, *p* = 0.27
SDQ—Peer difficulties	2.47 (2.26)	2 (1.76)	t = 0.66, *p* = 0.52
SDQ—Prosocial difficulties	7.93 (2.60)	8.32 (1.86)	t = −0.50, *p* = 0.62
Maternal age (years)	43.81 (5.6)	45.73 (5.41)	t = −1.03, *p* = 0.31
	Frequencies	Frequencies	
Maternal Lifetime Diagnosis	AN (14) BN (2)		
Maternal Education	Higher education (13) Up to A level (3)	Higher education (18) Up to A level (2)	X^2^(1) = 0.57
Maternal Ethnicity	White/Caucasian (15) Other (1)	White/Caucasian (17) Other (3)	X^2^(1) = 0.69

Means (and standard deviations) are reported for age, breast developmental stage, pubic hair developmental stage, WASI-full-scale IQ (FIQ), SDQ—total difficulties, SDQ—conduct difficulties, SDQ—hyperactivity difficulties, SDQ—peer difficulties, SDQ—prosocial difficulties, and maternal age. ^1^ Group comparisons are based on independent-sample Welsh’s *t*-tests and chi-squared tests.

**Table 2 brainsci-13-00099-t002:** Neurocognitive results.

Outcome	Controls	High-Risk ED	Crude Group Comparison	Adjusted Group Comparison ^+^
AST—Correct	92.56 (6.48)	94.65 (4.96)	F(1,34) = 1.13, *p* = 0.30, η^2^ = 0.03	F(1,32) = 1.54, *p* = 0.22, η^2^= 0.04
AST—Congruency cost	43.86 (59.75)	71.25 (38.49)	F(1,34) = 2.52, *p* = 0.12, η^2^ = 0.07	F(1,32) = 2.29, *p* = 0.14, η^2^ = 0.14
AST—Switching cost	159.95 (94.22)	240.86 (111.80)	F(1,34) = 5.55, ***p* = 0.02**, η^2^ = 0.14	F(1,32) = 5.53, ***p* = 0.02**, η^2^ = 0.12
AST—Mean latency	682.67 (143.88)	774.52 (135.37)	F(1,34) = 3.81, *p* = 0.06, η^2^ =0.10	F(1,32) = 3.80, *p =* 0.06, η^2^ =0.09
RVP—Mean latency	364.94 (90.76)	376.57 (88.12)	F(1,32) = 0.06, *p* = 0.81, η^2^ =0.001	F(1,32) = 0.15, *p* = 0.70, η^2^ = 0.004
SWM—Errors	28.20 (23.20)	26.94 (21.67)	F(1,34) = 0.03, *p* = 0.87, η^2^ =0.001	F(1,32) = 0.19, *p* = 0.67, η^2^ =0.003
SWM—Strategy	28.65 (9.01)	33.44 (6.69)	F(1,34) = 3.13, *p* = 0.09, η^2^ =0.08	F(1,32) = 2.96, *p =* 0.09, η^2^ = 0.08

Means (and standard deviations) are reported for each variable. Crude group comparison based on ANOVA. ^+^ = Adjusted group comparison based on ANCOVA correcting for age and FIQ. Bold = statistically significant difference (*p* < 0.05).

**Table 3 brainsci-13-00099-t003:** Spatial differences in functional connectivity within resting-state networks.

Regions	Hemisphere	T	n Voxels	*p*-Value (FDR Corrected)	x	y	z
**Posterior DMN (IC6): Controls > High-risk ED children**							
Angular Gyrus	Left	4.99	22	0.048	−38	−66	22
Middle Temporal Gyrus	Left	4.61	31	0.016	−40	−60	12
**Dorsal DMN (IC28): Controls > High-risk ED children**							
Medial Superior Frontal Gyrus	Right	5.8	52	<0.001	2	42	44
**Sensorimotor Network (IC30): Controls > High-risk ED children**							
Postcentral Gyrus	Left	4.7	25	0.012	−46	−10	34
**Attentional Network (IC9): Controls > High-risk ED children**							
Postcentral Gyrus	Left	5.78	32	0.03	−48	−18	62
Inferior Temporal Gyrus	Right	5.22	24	0.03	50	−64	−2
**Visual Network (IC1): High-risk ED children > Controls**							
Fusiform Gyrus	Right	4.79	21	0.028	22	−64	−12
Superior Occipital Gyrus	Left	4.6	29	0.024	−14	−100	26

## Data Availability

The data presented in this study are available on request from the corresponding author.

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
