# Peer review of "Neurocognitive Endophenotypes for Eating Disorders: A Preliminary High-Risk Family Study"

_brainsci, 2023, doi:10.3390/brainsci13010099_

Round 1
Reviewer 1 Report
Authors did not report comparisons between the two groups with respect to psychopathology. In Methods section is reported that authors assessed psychopathology using the Development and Wellbeing Assessment, but in Results section the comparison for DAWBA scores is missing. All the inferences mad by authors in Discussion section are valid only if the two groups did not show significant differences for ED and general psychopathology.
Author Response
We thank the reviewer for raising this important point. The full analysis of DAWBA scores is beyond the scope of this article, and is the subject of another publication.
As a screening tool for psychopathology in high-risk children and controls, we also used the Strenghth and Difficulties Questionnaire (SDQ). We have now reported in the text (and in Table 1) SDQ's scores and comparisons on each subscale of the questionnaire. Our analysis (reported in Table 1) shows that there were no between-group differences in any of the tool's subscales.
Reviewer 2 Report
The paper is interesting and well written. The limits of the paper are the sample size and to have included both AN and BN mothers of healthy daughters. These limits prevent to generalize the results of the study.
Besides, the authors themselves enlighten the limits of the study
Author Response
We thank the reviewer for the feedback. Although the sample size is limited, we have pointed out this limitation in the appropriate section.
Reviewer 3 Report
Dear Authors,
We consider article titled “Neurocognitive endophenotypes for eating disorders: a preliminary high-risk family study” a very interesting paper about the relevance of cognitive flexibility and functional connectivity as potential endophenotypes of Eating Disorder. In our opinion, this research conducted with an innovative study design offer a precious contribute to scientific community.
We are reporting some comments as follows in order to recommend some implementations:
- In Introduction section at line 79 we suggest to insert a reference that justify the reasons why only females are considered at higher risk for Eating Disorders and then included in the study.
- In Materials and Methods section we propose to specify if the study has been approved by Ethics Committee and if the subjects have given their informed consent.
- In Measures section at Sociodemographic data subgraph (line 113) the Authors could specify what questionnaire has been used (it is built at hoc questionnaire, or it is a standardized questionnaire?)
- We see that the definition of Default Mode Network is explained in Discussion section at line 320. Given the importance of this concept in the entire study, maybe the Authors could better explain the topic also in Introduction section, for a better understanding of its role by the lectors.
- In Discussion section in Strengths and Limitations subgraph, we suggest to include in limitations of the study that the sample consider only female subjects. We think that it could be interesting also consider the male population for future developments or to valuate gender differences in the phenomena.
Author Response
Dear Editors,
thank you for giving us the opportunity to revise the manuscript. Your precious feedback, combined with that of the reviewers, was very helpful in improving the text. We have highlighted in yellow the most important changes in the text. We are confident that this version of the manuscript may be suitable for publication.
We responded to all points arised in the review phase. Please, you can find our responses below.
Sincerely,
Edoardo Pappaianni, Nadia Micali, on behalf of the other co-authors.
We are reporting some comments as follows in order to recommend some implementations:
- In Introduction section at line 79 we suggest to insert a reference that justify the reasons why only females are considered at higher risk for Eating Disorders and then included in the study.
We thank the reviewers for highlighting this lack of clarity. We have added now a reference explaining how ED prevalence in women and men is different.
- In Materials and Methods section we propose to specify if the study has been approved by Ethics Committee and if the subjects have given their informed consent.
We've added now information about Ethics Committee approval and informed consent.
- In Measures section at Sociodemographic data subgraph (line 113) the Authors could specify what questionnaire has been used (it is built at hoc questionnaire, or it is a standardized questionnaire?)
We've specified now that the questionnaire was built ad-hoc.
- We see that the definition of Default Mode Network is explained in Discussion section at line 320. Given the importance of this concept in the entire study, maybe the Authors could better explain the topic also in Introduction section, for a better understanding of its role by the lectors.
We thank the reviewer for highlighting this. We have now spent a few more words about the Default-Mode Network in the Introduction, so that the lectors can understand better its fundamental role.
- In Discussion section in Strengths and Limitations subgraph, we suggest to include in limitations of the study that the sample consider only female subjects. We think that it could be interesting also consider the male population for future developments or to valuate gender differences in the phenomena.
We totally agree with the reviewers that it would be really interesting to study male population as well, culpably neglected in research so far. Hopefully this is going to change in the future. We've now added this important aspect as "future direction" in the appropriate section.